# Natural Value Approximators:
# Learning when to Trust Past Estimates

**Zhongwen Xu**
DeepMind
zhongwen@google.com

**Joseph Modayil**
DeepMind
modayil@google.com

**Hado van Hasselt**
DeepMind
hado@google.com

**Andre Barreto**
DeepMind
andrebarreto@google.com

**David Silver**
DeepMind
davidsilver@google.com

**Tom Schaul**
DeepMind
schaul@google.com

## Abstract

Neural networks have a smooth initial inductive bias, such that small changes in input do not lead to large changes in output. However, in reinforcement learning domains with sparse rewards, value functions have non-smooth structure with a characteristic asymmetric discontinuity whenever rewards arrive. We propose a mechanism that learns an interpolation between a direct value estimate and a projected value estimate computed from the encountered reward and the previous estimate. This reduces the need to learn about discontinuities, and thus improves the value function approximation. Furthermore, as the interpolation is learned and state-dependent, our method can deal with heterogeneous observability. We demonstrate that this one change leads to significant improvements on multiple Atari games, when applied to the state-of-the-art A3C algorithm.

## 1   Motivation

The central problem of reinforcement learning is value function approximation: how to accurately estimate the total future reward from a given state. Recent successes have used deep neural networks to approximate the value function, resulting in state-of-the-art performance in a variety of challenging domains [9]. Neural networks are most effective when the desired target function is smooth. However, value functions are, by their very nature, discontinuous functions with sharp variations over time. In this paper we introduce a representation of value that matches the natural temporal structure of value functions.

A value function represents the expected sum of future discounted rewards. If non-zero rewards occur infrequently but reliably, then an accurate prediction of the cumulative discounted reward rises as such rewarding moments approach and drops immediately after. This is depicted schematically with the dashed black line in Figure 1. The true value function is quite smooth, except immediately after receiving a reward when there is a sharp drop. This is a pervasive scenario because many domains associate positive or negative reinforcements to salient events (like picking up an object, hitting a wall, or reaching a goal position). The problem is that the agent's observations tend to be smooth in time, so learning an accurate value estimate near those sharp drops puts strain on the function approximator – especially when employing differentiable function approximators such as neural networks that naturally make smooth maps from observations to outputs.

To address this problem, we incorporate the temporal structure of cumulative discounted rewards into the value function itself. The main idea is that, by default, the value function can respect the reward sequence. If no reward is observed, then the next value smoothly matches the previous value, but

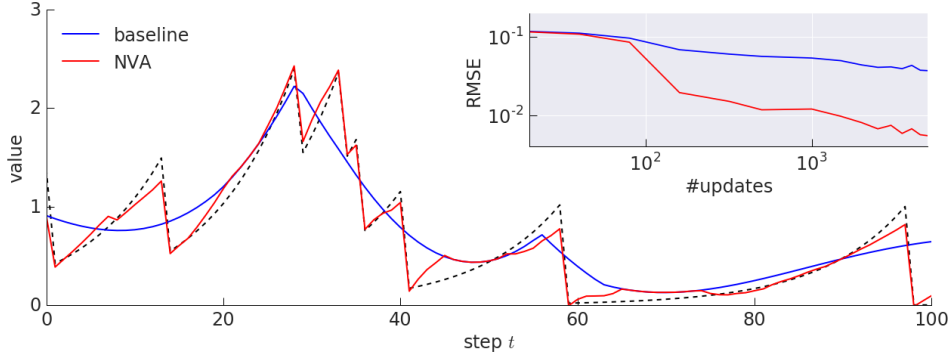

Figure 1: After the same amount of training, our proposed method (red) produces much more accurate estimates of the true value function (dashed black), compared to the baseline (blue). The main plot shows discounted future returns as a function of the step in a sequence of states; the inset plot shows the RMSE when training on this data, as a function of network updates. See section 4 for details.

becomes a little larger due to the discount. If a reward is observed, it should be subtracted out from the previous value: in other words a reward that was expected has now been consumed. The *natural value approximator* (NVA) combines the previous value with the observed rewards and discounts, which makes this sequence of values easy to represent by a smooth function approximator such as a neural network.

Natural value approximators may also be helpful in partially observed environments. Consider a situation in which an agent stands on a hill top. The goal is to predict, at each step, how many steps it will take until the agent has crossed a valley to another hill top in the distance. There is fog in the valley, which means that if the agent's state is a single observation from the valley it will not be able to accurately predict how many steps remain. In contrast, the value estimate from the initial hill top may be much better, because the observation is richer. This case is depicted schematically in Figure 2. Natural value approximators may be effective in these situations, since they represent the current value in terms of previous value estimates.

## 2 Problem definition

We consider the typical scenario studied in reinforcement learning, in which an agent interacts with an environment at discrete time intervals: at each time step $t$ the agent selects an action as a function of the current state, which results in a transition to the next state and a reward. The goal of the agent is to maximize the discounted sum of rewards collected in the long run from a set of initial states [12].

The interaction between the agent and the environment is modelled as a Markov Decision Process (MDP). An MDP is a tuple $(\mathcal{S}, \mathcal{A}, R, \gamma, P)$ where $\mathcal{S}$ is a state space, $\mathcal{A}$ is an action space, $R : \mathcal{S} \times \mathcal{A} \times \mathcal{S} \to \mathcal{D}(\mathbb{R})$ is a reward function that defines a distribution over the reals for each combination of state, action, and subsequent state, $P : \mathcal{S} \times \mathcal{A} \to \mathcal{D}(\mathcal{S})$ defines a distribution over subsequent states for each state and action, and $\gamma_t \in [0, 1]$ is a scalar, possibly time-dependent, discount factor.

One common goal is to make accurate predictions under a behaviour policy $\pi : \mathcal{S} \to \mathcal{D}(\mathcal{A})$ of the value

$$v_\pi(s) \equiv \mathbb{E}\left[R_1 + \gamma_1 R_2 + \gamma_1 \gamma_2 R_3 + \ldots \mid S_0 = s\right] . \tag{1}$$

The expectation is over the random variables $A_t \sim \pi(S_t)$, $S_{t+1} \sim P(S_t, A_t)$, and $R_{t+1} \sim R(S_t, A_t, S_{t+1})$, $\forall t \in \mathbb{N}^+$. For instance, the agent can repeatedly use these predictions to improve its policy. The values satisfy the recursive Bellman equation [2]

$$v_\pi(s) = \mathbb{E}\left[R_{t+1} + \gamma_{t+1} v_\pi(S_{t+1}) \mid S_t = s\right] .$$

We consider the common setting where the MDP is not known, and so the predictions must be learned from samples. The predictions made by an approximate value function $v(s; \boldsymbol{\theta})$, where $\boldsymbol{\theta}$ are parameters that are learned. The approximation of the true value function can be formed by temporal

difference (TD) learning [10], where the estimate at time $t$ is updated towards

$$Z_t^1 \equiv R_{t+1} + \gamma_{t+1} v(S_{t+1}; \boldsymbol{\theta}) \quad \text{or} \quad Z_t^n \equiv \sum_{i=1}^{n} (\Pi_{k=1}^{i-1} \gamma_{t+k}) R_{t+i} + (\Pi_{k=1}^{n} \gamma_{t+k}) v(S_{t+n}; \boldsymbol{\theta}) \,, \quad (2)$$

where $Z_t^n$ is the $n$-step *bootstrap target*, and the TD-error is $\delta_t^n \equiv Z_t^n - v(S_t; \boldsymbol{\theta})$.

## 3 Proposed solution: Natural value approximators

The conventional approach to value function approximation produces a value estimate from features associated with the current state. In states where the value approximation is poor, it can be better to rely more on a combination of the observed sequence of rewards and older but more reliable value estimates that are projected forward in time. Combining these estimates can potentially be more accurate than using one alone.

These ideas lead to an algorithm that produces three estimates of the value at time $t$. The first estimate, $V_t \equiv v(S_t; \boldsymbol{\theta})$, is a conventional value function estimate at time $t$. The second estimate,

$$G_t^p \equiv \frac{G_{t-1}^\beta - R_t}{\gamma_t} \qquad \text{if } \gamma_t > 0 \text{ and } t > 0 \,, \quad (3)$$

is a *projected* value estimate computed from the previous value estimate, the observed reward, and the observed discount for time $t$. The third estimate,

$$G_t^\beta \equiv \beta_t G_t^p + (1 - \beta_t) V_t = (1 - \beta_t) V_t + \beta_t \frac{G_{t-1}^\beta - R_t}{\gamma_t} \,, \quad (4)$$

is a convex combination of the first two estimates[1] formed by a time-dependent blending coefficient $\beta_t$. This coefficient is a learned function of state $\beta(\cdot; \boldsymbol{\theta}) : \mathcal{S} \to [0, 1]$, over the same parameters $\boldsymbol{\theta}$, and we denote $\beta_t \equiv \beta(S_t; \boldsymbol{\theta})$. We call $G_t^\beta$ the *natural value estimate* at time $t$ and we call the overall approach *natural value approximators* (NVA). Ideally, the natural value estimate will become more accurate than either of its constituents from training.

The value is learned by minimizing the sum of two losses. The first loss captures the difference between the conventional value estimate $V_t$ and the target $Z_t$, weighted by how much it is used in the natural value estimate,

$$J_V \equiv \mathbb{E}\left[ [\![ 1 - \beta_t ]\!] ([\![ Z_t ]\!] - V_t)^2 \right] \,, \quad (5)$$

where we introduce the stop-gradient identity function $[\![ x ]\!] = x$ that is defined to have a zero gradient everywhere, that is, gradients are not back-propagated through this function. The second loss captures the difference between the natural value estimate and the target, but it provides gradients only through the coefficient $\beta_t$,

$$J_\beta \equiv \mathbb{E}\left[ ([\![ Z_t ]\!] - (\beta_t [\![ G_t^p ]\!] + (1 - \beta_t) [\![ V_t ]\!]))^2 \right] \,. \quad (6)$$

These two losses are summed into a joint loss,

$$J = J_V + c_\beta J_\beta, \quad (7)$$

where $c_\beta$ is a scalar trade-off parameter. When conventional stochastic gradient descent is applied to minimize this loss, the parameters of $V_t$ are adapted with the first loss and parameters of $\beta_t$ are adapted with the second loss.

When bootstrapping on future values, the most accurate value estimate is best, so using $G_t^\beta$ instead of $V_t$ leads to refined prediction targets

$$Z_t^{\beta,1} \equiv R_{t+1} + \gamma_{t+1} G_{t+1}^\beta \quad \text{or} \quad Z_t^{\beta,n} \equiv \sum_{i=1}^{n} (\Pi_{k=1}^{i-1} \gamma_{t+k}) R_{t+i} + (\Pi_{k=1}^{n} \gamma_{t+k}) G_{t+n}^\beta \,. \quad (8)$$

## 4 Illustrative Examples

We now provide some examples of situations where natural value approximations are useful. In both examples, the value function is difficult to estimate well uniformly in all states we might care about, and the accuracy can be improved by using the natural value estimate $G_t^\beta$ instead of the direct value estimate $V_t$.

**Sparse rewards**  Figure 1 shows an example of value function approximation. To separate concerns, this is a supervised learning setup (regression) with the true value targets provided (dashed black line). Each point $0 \leq t \leq 100$ on the horizontal axis corresponds to one state $S_t$ in a single sequence. The shape of the target values stems from a handful of reward events, and discounting with $\gamma = 0.9$. We mimic observations that smoothly vary across time by 4 equally spaced radial basis functions, so $S_t \in \mathbb{R}^4$. The approximators $v(s)$ and $\beta(s)$ are two small neural networks with one hidden layer of 32 ReLU units each, and a single linear or sigmoid output unit, respectively. The input to $\beta$ is augmented with the last $k = 16$ rewards. For the baseline experiment, we fix $\beta_t = 0$. The networks are trained for 5000 steps using Adam [5] with minibatch size 32. Because of the small capacity of the $v$-network, the baseline struggles to make accurate predictions and instead it makes systematic errors that smooth over the characteristic peaks and drops in the value function. The natural value estimation obtains ten times lower root mean squared error (RMSE), and it also closely matches the qualitative shape of the target function.

**Heterogeneous observability**  Our approach is not limited to the sparse-reward setting. Imagine an agent that stands on the top of a hill. By looking in the distance, the agent may be able to predict how many steps should be taken to take it to the next hill top. When the agent starts descending the hill, it walks into fog in the valley between the hills. There, it can no longer see where it is. However, it could still determine how many steps until the next hill by using the estimate from the first hill and then simply counting steps. This is exactly what the natural value estimate $G_t^\beta$ will give us, assuming $\beta_t = 1$ on all steps in the fog. Figure 2 illustrates this example, where we assumed each step has a reward of $-1$ and the discount is one. The best observation-dependent value $v(S_t)$ is shown in dashed blue. In the fog, the agent can then do no better than to estimate the average number of steps from a foggy state until the next hill top. In contrast, the true value, shown in red, can be achieved exactly with natural value estimates. Note that in contrast to Figure 1, rewards are dense rather than sparse.

In both examples, we can sometimes trust past value functions more than current estimations, either because of function approximation error, as in the first example, or partial observability.

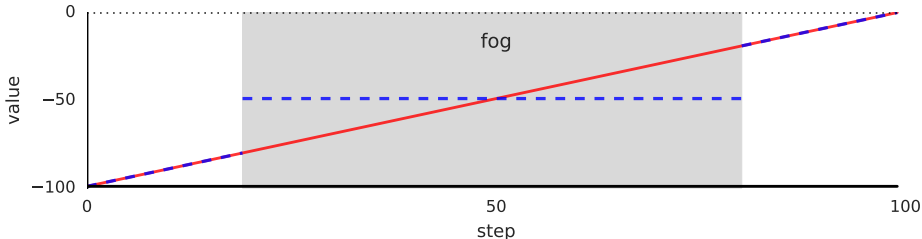

Figure 2: The value is the negative number of steps until reaching the destination at $t = 100$. In some parts of the state space, all states are aliased (in the fog). For these aliased states, the best estimate based only on immediate observations is a constant value (dashed blue line). Instead, if the agent relies on the value just before the fog and then decrements it by encountered rewards, while ignoring observations, then the agent can match the true value (solid red line).

## 5  Deep RL experiments

In this section, we integrate our method within A3C (Asynchronous advantage actor-critic [9]), a widely used deep RL agent architecture that uses a shared deep neural network to both estimate the policy $\pi$ (actor) and a baseline value estimate $v$ (critic). We modify it to use $G_t^\beta$ estimates instead of the regular value baseline $V_t$. In the simplest, feed-forward variant, the network architecture is composed of three layers of convolutions, followed by a fully connected layer with output $h$, which feeds into the two separate heads ($\pi$ with an additional softmax, and a scalar $v$, see the black components in the diagram below). The updates are done online with a buffer of the past 20-state transitions. The value targets are $n$-step targets $Z_t^n$ (equation 2) where each $n$ is chosen such that it bootstraps on the state at the end of the 20-state buffer. In addition, there is a loss contribution from the actor's policy gradient update on $\pi$. We refer the reader to [9] for details.

Table 1: Mean and median human-normalized scores on 57 Atari games, for the A3C baselines and our method, using both evaluation metrics. $N_{75}$ indicates the number of games that achieve at least 75% human performance.

| Agent | human starts | | | no-op starts | | |
|---|---|---|---|---|---|---|
| | $N_{75}$ | median | mean | $N_{75}$ | median | mean |
| A3C baseline | 28/57 | 68.5% | 310.4% | 31/57 | 91.6% | 334.0% |
| A3C + NVA | 30/57 | 93.5% | 373.3% | 32/57 | 117.0% | 408.4% |

Our method differs from the baseline A3C setup in the form of the value estimator in the critic ($G_t^\beta$ instead of $V_t$), the bootstrap targets ($Z_t^{\beta,n}$ instead of $Z_t^n$) and the value loss ($J$ instead of $J_V$) as discussed in section 3. The diagram on the right shows those new components in green; thick arrows denote functions with learnable parameters, thin ones without. In terms of the network architecture, we parametrize the blending coefficient $\beta$ as a linear function of the hidden representation $h$ concatenated with a window of past rewards $R_{t-k:t}$ followed by a sigmoid:

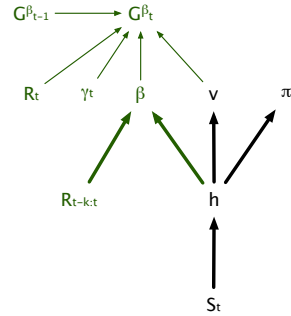

$$\beta(S_t; \boldsymbol{\theta}) \equiv \frac{\gamma_t}{1 + \exp\left(\boldsymbol{\theta}_\beta^\top [h(S_t); R_{t-k:t}]\right)}, \qquad (9)$$

where $\boldsymbol{\theta}_\beta$ are the parameters of the $\beta$ head of the network, and we set $k$ to 50. The extra factor of $\gamma_t$ handles the otherwise undefined beginnings of episode (when $\gamma_0 = 0$), and it ensures that the time-scale across which estimates can be projected forward cannot exceed the time-scale induced by the discounting[2].

We investigate the performance of natural value estimates on a collection of 57 video games games from the Atari Learning Environment [1], which has become a standard benchmark for Deep RL methods because of the rich diversity of challenges present in the various games. We train agents for 80 Million agent steps (320 Million Atari game frames) on a single machine with 16 cores, which corresponds to the number of frames denoted as '1 day on CPU' in the original A3C paper. All agents are run with one seed and a single, fixed set of hyper-parameters. Following [8], the performance of the final policy is evaluated under two modes, with a random number of no-ops at the start of each episode, and from randomized starting points taken from human trajectories.

## 5.1 Results

Table 1 summarizes the aggregate performance results across all 57 games, normalized by human performance. The evaluation results are presented under two different conditions, the human starts condition evaluates generalization to a different starting state distribution than the one used in training, and the no-op starts condition evaluates performance on the same starting state distribution that was used in training. We summarize normalized performance improvements in Figure 3. In the appendix, we provide full results for each game in Table 2 and Table 3. Across the board, we find that adding NVA improves the performance on a number of games, and improves the median normalized score by 25% or 25.4% for the respective evaluation metrics.

The second measure of interest is the change in value error when using natural value estimates; this is shown in Figure 4. The summary across all games is that the the natural value estimates are more accurate, sometimes substantially so. Figure 4 also shows detailed plots from a few representative games, showing that large accuracy gaps between $V_t$ and $G^\beta$ lead to the learning of larger blending proportions $\beta$.

The fact that more accurate value estimates improve final performance on only some games should not be surprising, as they only directly affect the critic and they affect the actor indirectly. It is also

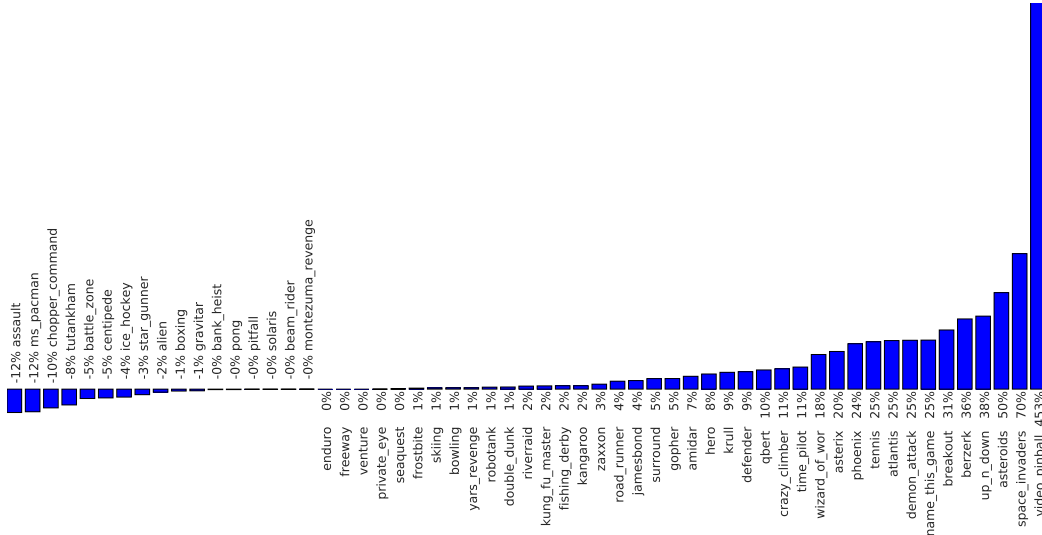

Figure 3: The performance gains of the proposed architecture over the baseline system, with the performance normalized for each game with the formula $\frac{proposed-baseline}{\max(human,baseline)-random}$ used previously in the literature [15].

unclear for how many games the bottleneck is value accuracy instead of exploration, memory, local optima, or sample efficiency.

## 6 Variants

We explored a number of related variants on the subset of tuning games, with mostly negative results, and report our findings here, with the aim of adding some additional insight into what makes NVA work—and to prevent follow-up efforts from blindly repeating our mistakes.

$\beta$**-capacity** We experimented with adding additional capacity to the $\beta$-network in Equation 9, namely inserting a hidden ReLU layer with $n_h \in \{16, 32, 64\}$; this neither helped nor hurt performance, so opted for the simplest architecture (no hidden layer). We hypothesize that learning a binary gate is much easier than learning the value estimate, so no additional capacity is required.

**Weighted $v$-updates** We also validated the design choice of weighting the update to $v$ by its usage $(1 - \beta)$ (see Equation 5). On the 6 tuning games, weighting by usage obtains slightly higher performance than an unweighted loss on $v$. One hypothesis is that the weighting permits the direct estimates to be more accurate in some states than in others, freeing up function approximation capacity for where it is most needed.

**Semantic versus aggregate losses** Our proposed method separates the *semantically* different updates on $\beta$ and $v$, but of course a simpler alternative would be to directly regress the natural value estimate $G_t^\beta$ toward its target, and back-propagate the *aggregate* loss into both $\beta$ and $v$ jointly. This alternative performs substantially worse, empirically. We hypothesize one reason for this: in a state where $G_t^p$ structurally over-estimates the target value, an aggregate loss will encourage $v$ to compensate by under-estimating it. In contrast, the semantic losses encourage $v$ to simply be more accurate and then reduce $\beta$.

**Training by back-propagation through time** The recursive form of Equation 4 lends itself to an implementation as a specific form of recurrent neural network, where the recurrent connection transmits a single scalar $G_t^\beta$. In this form, the system can be trained by back-propagation through time (BPTT [17]). This is semantically subtly different from our proposed method, as the gates $\beta$ no longer make a local choice between $V_t$ and $G_t^p$, but instead the entire sequence of $\beta_{t-k}$ to $\beta_t$ is

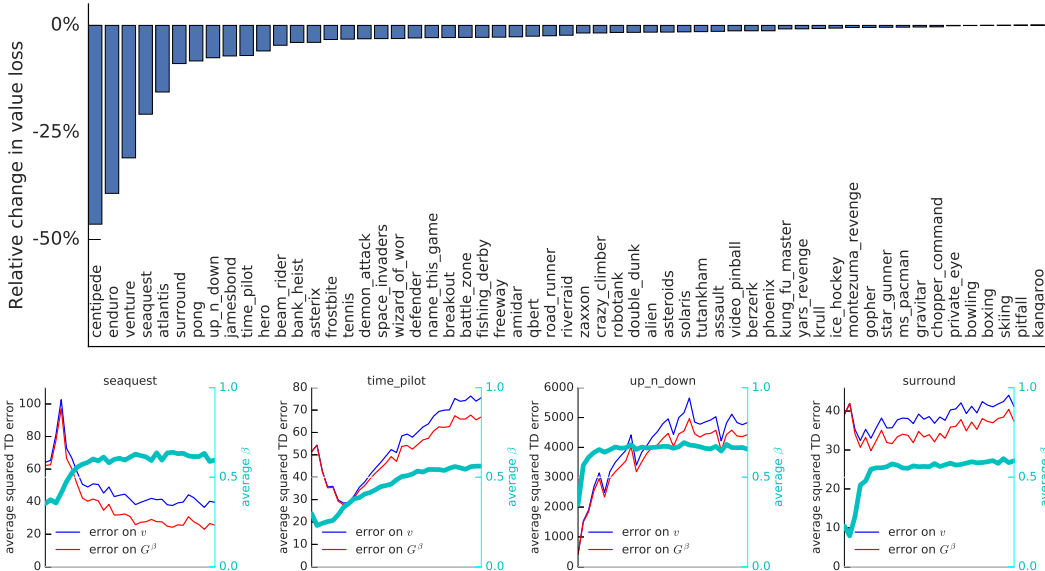

Figure 4: Reduction in value estimation error compared to the baseline. The proxies we use are average squared TD-errors encountered during training, comparing $\epsilon_v = \frac{1}{2}(Z_t - v(S_t; \boldsymbol{\theta}))^2$ and $\epsilon_\beta = \frac{1}{2}(Z_t - G_t^\beta)^2$. **Top**: Summary graph for all games, showing relative change in error $(\epsilon_\beta - \epsilon_v)/\epsilon_v$, averaged over the full training run. As expected, the natural value estimate consistently has equal or lower error, validating our core hypothesis. **Bottom:** Detailed plots on a handful of games. It shows the direct estimate error $\epsilon_v$ (blue) and natural value estimate error $\epsilon_\beta$ (red). In addition, the blending proportion $\beta$ (cyan) adapts over time to use more of the prospective value estimate if that is more accurate.

trained to provide the best estimate $G_t^\beta$ at time $t$ (where $k$ is the truncation horizon of BPTT). We experimented with this variant as well: it led to a clear improvement over the baseline as well, but its performance was substantially below the simpler feed-forward setup with reward buffer in Equation 9 (median normalized scores of 78% and 103% for the human and no-op starts respectively).

## 7 Discussion

**Relation to eligibility traces**    In TD($\lambda$) [11], a well-known and successful variant of TD, the value function (1) is not learned by a one-step update, but instead relies on multiple value estimates from further in the future. Concretely, the target for the update of the estimate $V_t$ is then $G_t^\lambda$, which can be defined recursively by $G_t^\lambda = R_{t+1} + \gamma_{t+1}(1 - \lambda)V_{t+1} + \gamma_{t+1}\lambda G_{t+1}^\lambda$, or as a mixture of several $n$-step targets [12]. The trace parameter $\lambda$ is similar to our $\beta$ parameter, but faces backwards in time rather than forwards.

A quantity very similar to $G_t^\beta$ was discussed by van Hasselt and Sutton [13], where this quantity was then used to update values prior to time $t$. The inspiration was similar, in the sense that it was acknowledged that $G_t^\beta$ may be a more accurate target to use than either the Monte Carlo return or any single estimated state value. The use of $G_t^\beta$ itself for online predictions, apart from using it as a target to update towards, was not yet investigated.

**Extension to action-values**    There is no obstacle to extend our approach to estimators of action-values $q(S_t, A_t, \boldsymbol{\theta})$. One generalization from TD to SARSA is almost trivial. The quantity $G_t^\beta$ then has the semantics of the value of action $A_t$ in state $S_t$.

It is also possible to consider off-policy learning. Consider the Bellman optimality equation $Q^*(s, a) = \mathbb{E}\left[R_{t+1} + \gamma_{t+1} \max_{a'} Q^*(S_{t+1}, a')\right]$. This implies that for the optimal value function $Q^*$,

$$\mathbb{E}\left[\max_a Q^*(S_t, a)\right] = \mathbb{E}\left[\frac{Q^*(S_{t-1}, A_{t-1}) - R_t}{\gamma_t}\right].$$

This implies that we may be able to use the quantity $(Q(S_{t-1}, A_{t-1}) - R_t)/\gamma_t$ as an estimate for the greedy value $\max_a Q(S_t, a)$. For instance, we could blend the value as in SARSA, and define

$$G_t^\beta = (1 - \beta_t)Q(S_t, A_t) + \beta_t \frac{G_{t-1}^\beta - R_t}{\gamma_t} \,.$$

Perhaps we could require $\beta_t = 0$ whenever $A_t \neq \arg\max_a Q(S_t, a)$, in a similar vein as Watkins' $Q(\lambda)$ [16] that zeros the eligibility trace for non-greedy actions. We leave this and other potential variants for more detailed consideration in future work.

**Memory**   NVA adds a small amount of memory to the system (a single scalar), which raises the question of whether other forms of memory, such as the LSTM [4], provide a similar benefit. We do not have a conclusive answer, but the existing empirical evidence indicates that the benefit of natural value estimation goes beyond just memory. This can be seen by comparing to the A3C+LSTM baseline (also proposed in [9]), which has vastly larger memory and number of parameters, yet did not achieve equivalent performance (median normalized scores of 81% for the human starts). To some extent this may be caused by the fact that recurrent neural networks are more difficult to optimize.

**Regularity and structure**   Results from the supervised learning literature indicate that computing a reasonable approximation of a given target function is feasible when the learning algorithm exploits some kind of regularity in the latter [3]. For example, one may assume that the target function is bounded, smooth, or lies in a low-dimensional manifold. These assumptions are usually materialised in the choice of approximator. Making structural assumptions about the function to approximate is both a blessing and a curse. While a structural assumption makes it possible to compute an approximation with a reasonable amount of data, or using a smaller number of parameters, it can also compromise the quality of the solution from the outset. We believe that while our method may not be the ideal structural assumption for the problem of approximating value functions, it is at least *better* than the smooth default.

**Online learning**   By construction, the natural value estimates are an online quantity, that can only be computed from a trajectory. This means that the extension to experience replay [6] is not immediately obvious. It may be possible to replay trajectories, rather than individual transitions, or perhaps it suffices to use stale value estimates at previous states, which might still be of better quality than the current value estimate at the sampled state. We leave a full investigation of the combination of these methods to future work.

**Predictions as state**   In our proposed method the value is estimated in part as a function of a single past prediction, and this has some similarity to past work in predictive state representations [7]. Predictive state representations are quite different in practice: their state consists of only predictions, the predictions are of future observations and actions (not rewards), and their objective is to provide a sufficient representation of the full environmental dynamics. The similarities are not too strong with the work proposed here, as we use a single prediction of the actual value, this prediction is used as a small but important part of the state, and the objective is to estimate only the value function.

## 8   Conclusion

This paper argues that there is one specific structural regularity that underlies the value function of many reinforcement learning problems, which arises from the temporal nature of the problem. We proposed natural value approximation, a method that learns how to combine a direct value estimate with ones projected from past estimates. It is effective and simple to implement, which we demonstrated by augmenting the value critic in A3C, and which significantly improved median performance across 57 Atari games.

**Acknowledgements**

The authors would like to thank Volodymyr Mnih for his suggestions and comments on the early version of the paper, the anonymous reviewers for constructive suggestions to improve the paper. The authors also thank the DeepMind team for setting up the environments and building helpful tools used in the paper.

## Footnotes

[1]Note the mixed recursion in the definition, $G^p$ depends on $G^\beta$, and vice-versa.

[2]This design choice may not be ideal in all circumstances, sometimes projecting old estimates further can perform better—our variant however has the useful side-effect that the weight for the $V_t$ update (Equation 5) is now greater than zero independently of $\beta$. This prevents one type of vicious cycle, where an initially inaccurate $V_t$ leads to a large $\beta$, which in turn reduces the learning of $V_t$, and leads to an unrecoverable situation.

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
