[Supplementary Material]

# A Hyper-parameters

To determine good hyper-parameters, we evaluated performance on six games (Beam Rider, Breakout, Pong, Q-bert, Space Invaders, Seaquest, as in [14]). We varied learning rates, sampling from $LogUniform(10^{-4}, 10^{-2})$, and found the learning rate to work robustly between $5 \times 10^{-4}$ to $10^{-3}$. We thus opted to retain the canonical A3C baseline settings of learning rate $= 6 \times 10^{-4}$, entropy cost $= 10^{-2}$, unroll length $N = 20$, and RMSProp as optimizer [9]. We varied the cost trade-off $c_\beta$, sampling from $LogUniform(10^{-2}, 10^2)$, and found it to be robust between $0.1$ and $1$; we set it to $c_\beta = 0.5$ in all experiments. The final set of hyper-parameters was fixed across all 57 Atari games.

| Game | Random | Human | Baseline A3C | A3C+NVA |
|------|-------:|------:|-------------:|--------:|
| alien | 128.3 | 6,371.3 | 562.5 | 610.2 |
| amidar | 11.8 | 1,540.4 | 200.1 | 220.3 |
| assault | 166.9 | 628.9 | 3,275.8 | 2,908.3 |
| asterix | 164.5 | 7,536.0 | 7,123.8 | 7,718.3 |
| asteroids | 871.3 | 36,517.3 | 2,674.1 | 21,074.7 |
| atlantis | 13,463.0 | 26,575.0 | 710,499.5 | 882,379.0 |
| bank heist | 21.7 | 644.5 | 934.1 | 925.9 |
| battle zone | 3,560.0 | 33,030.0 | 10,710.0 | 9,715.0 |
| beam rider | 254.6 | 14,961.0 | 9,161.0 | 8,901.1 |
| berzerk | 196.1 | 2,237.5 | 763.1 | 1,417.7 |
| bowling | 35.2 | 146.5 | 46.1 | 47.2 |
| boxing | -1.5 | 9.6 | 71.2 | 54.0 |
| breakout | 1.6 | 27.9 | 423.0 | 563.9 |
| centipede | 1,925.5 | 10,321.9 | 4,094.1 | 1,893.4 |
| chopper command | 644.0 | 8,930.0 | 4,446.0 | 4,727.5 |
| crazy climber | 9,337.0 | 32,667.0 | 105,399.0 | 107,951.0 |
| defender | 1,965.5 | 14,296.0 | 39,208.3 | 41,940.8 |
| demon attack | 208.3 | 3,442.8 | 75,830.9 | 88,609.3 |
| double dunk | -16.0 | -14.4 | -0.2 | -0.1 |
| enduro | -81.8 | 740.2 | -82.3 | -82.5 |
| fishing derby | -77.1 | 5.1 | 15.1 | 18.5 |
| freeway | 0.1 | 25.6 | 0.1 | 0.1 |
| frostbite | 66.4 | 4,202.8 | 208.9 | 207.9 |
| gopher | 250.0 | 2,311.0 | 9,114.8 | 8,791.3 |
| gravitar | 245.5 | 3,116.0 | 310.8 | 292.3 |
| hero | 1,580.3 | 25,839.4 | 28,931.2 | 31,572.7 |
| ice hockey | -9.7 | 0.5 | -4.2 | -4.4 |
| jamesbond | 33.5 | 368.5 | 375.3 | 360.0 |
| kangaroo | 100.0 | 2,739.0 | 127.0 | 181.0 |
| krull | 1,151.9 | 2,109.1 | 5,455.6 | 5,783.4 |
| kung fu master | 304.0 | 20,786.8 | 28,954.8 | 27,917.0 |
| montezuma revenge | 25.0 | 4,182.0 | 49.0 | 51.5 |
| ms pacman | 197.8 | 15,375.0 | 996.9 | 896.0 |
| name this game | 1,747.8 | 6,796.0 | 5,204.6 | 6,193.5 |
| phoenix | 1,134.4 | 6,686.2 | 6,159.1 | 7,656.5 |
| pitfall | -348.8 | 5,998.9 | -188.7 | -137.8 |
| pong | -18.0 | 15.5 | 18.8 | 18.7 |
| private eye | 662.8 | 64,169.1 | 293.5 | 164.1 |
| qbert | 183.0 | 12,085.0 | 10,592.0 | 11,313.9 |
| riverraid | 588.3 | 14,382.2 | 9,569.0 | 9,516.1 |
| road runner | 200.0 | 6,878.0 | 37,313.5 | 38,147.0 |
| robotank | 2.4 | 8.9 | 2.3 | 2.3 |
| seaquest | 215.5 | 40,425.8 | 1,959.5 | 2,394.9 |
| skiing | -15,287.4 | -3,686.6 | -14,164.0 | -13,957.7 |
| solaris | 2,047.2 | 11,032.6 | 1,882.2 | 1,627.9 |
| space invaders | 182.6 | 1,464.9 | 866.2 | 1,833.0 |
| star gunner | 697.0 | 9,528.0 | 58,156.0 | 56,795.5 |
| surround | -9.7 | 5.4 | -8.2 | -7.0 |
| tennis | -21.4 | -6.7 | -8.0 | -7.1 |
| time pilot | 3,273.0 | 5,650.0 | 9,409.5 | 10,180.0 |
| tutankham | 12.7 | 138.3 | 159.4 | 138.3 |
| up n down | 707.2 | 9,896.1 | 83,976.3 | 97,562.4 |
| venture | 18.0 | 1,039.0 | 22.0 | 18.0 |
| video pinball | 0.0 | 15,641.1 | 30,912.4 | 210,555.7 |
| wizard of wor | 804.0 | 4,556.0 | 4,628.0 | 4,438.0 |
| yars revenge | 1,476.9 | 47,135.2 | 7,157.5 | 6,306.7 |
| zaxxon | 475.0 | 8,443.0 | 11,282.5 | 11,709.0 |

Table 2: Performance on all games for the human starts condition.

| Game | Random | Human | Baseline A3C | A3C+NVA |
|---|---|---|---|---|
| alien | 227.8 | 7,127.7 | 1,684.7 | 1,576.3 |
| amidar | 5.8 | 1,719.5 | 907.4 | 1,021.5 |
| assault | 222.4 | 742.0 | 3,484.9 | 3,089.4 |
| asterix | 210.0 | 8,503.3 | 16,364.0 | 19,518.0 |
| asteroids | 719.1 | 47,388.7 | 3,900.1 | 27,242.1 |
| atlantis | 12,850.0 | 29,028.1 | 711,682.5 | 887,543.0 |
| bank heist | 14.2 | 753.1 | 1,286.7 | 1,283.5 |
| battle zone | 2,360.0 | 37,187.5 | 13,745.0 | 12,050.0 |
| beam rider | 363.9 | 16,926.5 | 7,853.0 | 7,840.4 |
| berzerk | 123.7 | 2,630.4 | 892.5 | 1,804.7 |
| bowling | 23.1 | 160.7 | 36.9 | 37.9 |
| boxing | 0.1 | 12.1 | 90.7 | 89.8 |
| breakout | 1.7 | 30.5 | 466.1 | 608.5 |
| centipede | 2,090.9 | 12,017.0 | 5,476.5 | 5,025.3 |
| chopper command | 811.0 | 7,387.8 | 6,833.0 | 6,194.0 |
| crazy climber | 10,780.5 | 35,829.4 | 116,127.0 | 127,244.0 |
| defender | 2,874.5 | 18,688.9 | 45,526.0 | 49,383.3 |
| demon attack | 152.1 | 1,971.0 | 68,416.4 | 85,708.1 |
| double dunk | -18.6 | -16.4 | -0.6 | -0.4 |
| enduro | 0.0 | 860.5 | 0.0 | 0.0 |
| fishing derby | -91.7 | -38.7 | 32.9 | 35.1 |
| freeway | 0.0 | 29.6 | 0.0 | 0.0 |
| frostbite | 65.2 | 4,334.7 | 286.8 | 309.0 |
| gopher | 257.6 | 2,412.5 | 9,320.4 | 9,818.0 |
| gravitar | 173.0 | 3,351.4 | 246.8 | 219.3 |
| hero | 1,027.0 | 30,826.4 | 34,285.7 | 36,897.2 |
| ice hockey | -11.2 | 0.9 | -4.5 | -5.0 |
| jamesbond | 29.0 | 302.8 | 446.5 | 465.0 |
| kangaroo | 52.0 | 3,035.0 | 53.0 | 106.0 |
| krull | 1,598.0 | 2,665.5 | 6,765.6 | 7,216.1 |
| kung fu master | 258.5 | 22,736.3 | 34,471.0 | 35,051.5 |
| montezuma revenge | 0.0 | 4,753.3 | 0.5 | 0.0 |
| ms pacman | 307.3 | 6,951.6 | 3,458.5 | 2,679.5 |
| name this game | 2,292.3 | 8,049.0 | 6,006.9 | 7,469.8 |
| phoenix | 761.4 | 7,242.6 | 7,302.4 | 8,842.0 |
| pitfall | -229.4 | 6,463.7 | -4.7 | -10.9 |
| pong | -20.7 | 14.6 | 20.7 | 20.6 |
| private eye | 24.9 | 69,571.3 | 97.8 | 99.3 |
| qbert | 163.9 | 13,455.0 | 22,869.8 | 25,128.1 |
| riverraid | 1,338.5 | 17,118.0 | 14,330.1 | 14,580.1 |
| road runner | 11.5 | 7,845.0 | 44,329.0 | 46,152.5 |
| robotank | 2.2 | 11.9 | 2.7 | 2.8 |
| seaquest | 68.4 | 42,054.7 | 1,648.6 | 1,716.6 |
| skiing | -17,098.1 | -4,336.9 | -16,156.8 | -16,071.6 |
| solaris | 1,236.3 | 12,326.7 | 2,328.2 | 2,318.5 |
| space invaders | 148.0 | 1,668.7 | 1,294.2 | 2,362.1 |
| star gunner | 664.0 | 10,250.0 | 61,219.5 | 59,481.0 |
| surround | -10.0 | 6.5 | -7.7 | -6.8 |
| tennis | -23.8 | -8.3 | -6.3 | -2.0 |
| time pilot | 3,568.0 | 5,229.2 | 10,784.5 | 11,610.0 |
| tutankham | 11.4 | 167.6 | 300.9 | 277.2 |
| up n down | 533.4 | 11,693.2 | 114,094.1 | 157,066.1 |
| venture | 0.0 | 1,187.5 | 0.0 | 0.0 |
| video pinball | 0.0 | 17,667.9 | 35,435.7 | 195,883.1 |
| wizard of wor | 563.5 | 4,756.5 | 5,215.5 | 6,051.0 |
| yars revenge | 3,092.9 | 54,576.9 | 10,932.8 | 11,310.1 |
| zaxxon | 32.5 | 9,173.3 | 14,143.5 | 14,507.5 |

Table 3: Performance on all games for the no-op starts condition.