[Reviews · NeurIPS 2017]

Reviewer 1



This paper proposes a novel way to estimate the value function of a game state, by treating its previous reward and value estimations as additional input besides the current state. These additional input is the direct reason why value function depicts a non-smoothness structure (e.g., a sparse immediate rewards lead to a bump in the value function). By taking them into consideration explicitly, the value function can be more easily estimated. Although the proposed idea is quite interesting, there are a lot of baselines that the proposed method might need to compare. As mentioned in Section 7, eligible trace estimates the cumulative return as a linear combination of k-step *future* returns with geometric weights. Also Generalized Advantage Estimation, as a novel way to estimate the overall return using future estimation, is another example. Comparing these previous approaches with the proposed method that uses past estimation is both interesting and important. In addition, there exist many simple approaches that also captures the history (and sparse reward that happened recently), e.g., frame stacking. A comparison will also be interesting. The paper starts with the story that the value function is smooth, if the non-smooth part is explained away by previous rewards and values. There are also motivating examples (e.g., Fig. 1 and Fig. 2). However, there is no experiments in Atari games showing that the estimated value function using proposed approach is indeed smoother than the baseline. Table. 1 shows that while median score shows a strong boost, mean score of the proposed approach is comparable to the baseline. This might suggests that the proposed approach does a good job in reducing the variance of the trained models, rather than giving higher performance. In the paper, there seems to be no relevant analysis. Some detailed questions: 1. According to the paper, the modified A3C algorithm uses N = 20 steps, rather than N = 5 as in the vanilla A3C. Did the baseline also use N = 20? Note that N could be an important factor for performance, since with long horizon the algorithm will see more rewards in one gradient update given the same number of game frames. 2. Line 154 says "all agents are run with one seed". Does that mean the agents are initialized by the same random seed among different games, or the game environment starts from the same seed? Please clarify.

Reviewer 2



[I have read the other reviews and the author feedback. I maintain my rating that this paper should be accepted.] The paper proposes a novel way to blend "projected" TD errors into value function approximations and demonstrates downstream benefit in deep RL applications. Some of the claims are over-reaching (e.g. statements like "structure ... has never been exploited to obtain better value function approximations" lead the reader down the wrong path in context of van Hasselt and Sutton [13] ) and the paper may benefit from explaining connections to eligibility traces, PSR and [13] much earlier. That said, the core idea is simple, effective and easy to implement. Many questions I had (e.g. identifiability concerns between beta and v were heuristically handled by the stop-gradient operations in Eqn 5 and 6, perhaps an alternative way to do aggregate end-to-end training exists) were addressed in Section 6. At present, Figure 4 does not add value in my opinion. I'm not convinced that E_v and E_beta are comparable (the regression targets Z_t depend on many confounding factors). I'd rather see an exploration of the limits of the proposed approach: e.g. intuitively it seems that natural value estimates could "oscillate" and present challenges for reliable convergence. Minor: 128: instead *of* the 134: actor's policy gradient 151: because *of* the rich 180: instead of exploration 215: *be* defined recursively 220: acknowledged that 233: detailed consideration

Reviewer 3



Review of submission 1276: Natural value approximators: learning when to trust past estimates Summary: The Nature Value Approximators (NVA) utilizes the regularity in the value functions of reinforcement learning approaches to make more accurate value predictions. NVA learns to mix direct value estimates and past estimates automatically. Empirical results on ATARI games show NVA reduces the value prediction error substantially. The paper is very well written and was generally a pleasure to read. Comments: + To the best of my knowledge, NVA is the first one leveraging the regularity of value functions in RL to learn to leverage past estimates to improve value prediction automatically. The idea of NVA is novel and interesting. + NVA is essentially a low capacity memory mechanism. Memory mechanism has been investigated in RL to address partially observability and/or to reuse knowledge/skills in the setting of transfer learning/lifelong learning. It is interesting to see that memory can indeed help prediction beyond partial observability + Section 6 and 7 provides extensive discussion on NVA variants and related topics. They are informative and useful. - If we compare Figure 4 (Top) with Figure 3, the game Centipede has the greatest improvement on the value loss, but the worst performance gain. What property of the game Centipede caused this result?